# DHA: End-to-End Joint Optimization of Data Augmentation Policy, Hyper-parameter and Architecture

**Kaichen Zhou**
*University of Oxford*

**Lanqing Hong**[†]                                                        *honglanqing@huawei.com*
*Huawei Noah's Ark Lab*

**Shoukang Hu**
*The Chinese University of Hong Kong*

**Fengwei Zhou**
*Huawei Noah's Ark Lab*

**Binxin Ru**
*SailYond Technology & Research Institute of Tsinghua University in Shenzhen*

**Jiashi Feng**
*National University of Singapore*

**Zhenguo Li**
*Huawei Noah's Ark Lab*

**Reviewed on OpenReview:** *https://openreview.net/forum?id=MHOAEiTlen*

## Abstract

Automated machine learning (AutoML) usually involves several crucial components, such as Data Augmentation (DA) policy, Hyper-Parameter Optimization (HPO), and Neural Architecture Search (NAS). Although many strategies have been developed for automating these components in separation, joint optimization of these components remains challenging due to the largely increased search dimension and the variant input types of each component. In parallel to this, the common practice of *searching* for the optimal architecture first and then *retraining* it before deployment in NAS often suffers from low performance correlation between the searching and retraining stages. An end-to-end solution that integrates the AutoML components and returns a ready-to-use model at the end of the search is desirable. In view of these, we propose **DHA**, which achieves joint optimization of **D**ata augmentation policy, **H**yper-parameter and **A**rchitecture. Specifically, end-to-end NAS is achieved in a differentiable manner by optimizing a compressed lower-dimensional feature space, while DA policy and HPO are regarded as dynamic schedulers, which adapt themselves to the update of network parameters and network architecture at the same time. Experiments show that DHA achieves state-of-the-art (SOTA) results on various datasets and search spaces. To the best of our knowledge, we are the first to efficiently and jointly optimize DA policy, NAS, and HPO in an end-to-end manner without retraining.

## 1 Introduction

While deep learning has achieved remarkable progress in various tasks such as computer vision and natural language processing, the design and training of a well-performing deep neural architecture for a specific task usually requires tremendous human involvement (He et al., 2016; Sandler et al., 2018). To alleviate such

burden on human users, AutoML algorithms have been proposed in recent years to automate the pipeline of designing and training a model, such as automated Data Augmentation (DA), Hyper-Parameter Optimization (HPO), and Neural Architecture Search (NAS) (Cubuk et al., 2018; Mittal et al., 2020; Chen et al., 2019). All of these AutoML components are normally processed independently and the naive solution of applying them sequentially in separate stages, not only suffers from low efficiency but also leads to sub-optimal results (Dai et al., 2020; Dong et al., 2020). Indeed, how to achieve full-pipeline "from data to model" automation efficiently and effectively is still a challenging and open problem.

One of the main difficulties lies in understanding how to automatically combine the different AutoML components (e.g., NAS and HPO) appropriately without human expertise. FBNetV3 (Dai et al., 2020) and AutoHAS (Dong et al., 2020) investigated the joint optimization of NAS and HPO, while (Kashima et al., 2020) focused on the joint optimization of neural architectures and data augmentation policies. The joint optimization of NAS and quantization policy were also investigated in APQ (Wang et al., 2020). Clear benefits can be seen in the above works when optimizing two AutoML components together, which motivates the further investigation of "from data to model" automation. However, with the increasing number of AutoML components, the search space complexity is increased by several orders of magnitudes and it is challenging to operate in such a large search space. In addition, how these AutoML components affect each other when optimized together is still unclear. Thus, further investigation is needed to open the black box of optimizing different AutoML components jointly.

Another main challenge of achieving the automated pipeline "from data to model" is understanding how to perform end-to-end searching and training of models without the need of parameter retraining. Current approaches, even those considering only one AutoML component such as NAS algorithms, usually require two stages, one for searching and one for retraining (Liu et al., 2019; Xie et al., 2019). Similarly, automatic DA methods such as FastAA (Lim et al., 2019) also need to retrain the model parameters once the DA policies have been searched. In these cases, whether the searched architectures or DA policies would perform well after retraining is questionable, due to the inevitable difference of training setup between the searching and retraining stages (Yang et al., 2019). To improve the performance correlation between searching and retraining stages, DSNAS (Hu et al., 2020) developed a differentiable NAS method to provide direct NAS without parameter retraining. OnlineAugment (Tang et al., 2020) and OnlineHPO (Im et al., 2021) design direct DA or HPO policy, respectively, without model retraining.

Targeting the challenging task-specific end-to-end AutoML, we propose DHA, a differentiable joint optimization solution for efficient end-to-end AutoML components, including DA, HPO and NAS. Specifically, the DA and HPO are regarded as dynamic schedulers, which adapt themselves to the update of network parameters and network architecture. At the same time, the end-to-end NAS optimization is realized in a differentiable manner with the help of sparse coding method, which means that instead of performing our search in a high-dimensional network architecture space, we optimize a compressed lower-dimensional feature space. With this differentiable manner, DHA can effectively deal with the huge search space and the high optimization complexity caused by the joint optimization problem. To summarize, our main contributions are as follows:

- We propose an AutoML method, DHA, for the concurrent optimization of DA, HPO, and NAS. To the best of our knowledge, we are the first to efficiently and jointly realize DA, HPO, and NAS in an end-to-end manner without retraining.

- Experiments show that DHA achieves a state-of-the-art (SOTA) accuracy on ImageNet with both cell-based and Mobilenet-like architecture search space. DHA also provides SOTA results on various datasets, including CIFAR10, CIFAR100, SPORT8, MIT67, FLOWERS102 and ImageNet with relatively low computational cost, showing the effectiveness and efficiency of joint optimization (see Fig. 2).

- Through extensive experiments, we demonstrate the advantages of doing joint-training over optimizing each AutoML component in sequence. Besides, higher model performance and a smoother loss landscape are achieved by our proposed DHA method.

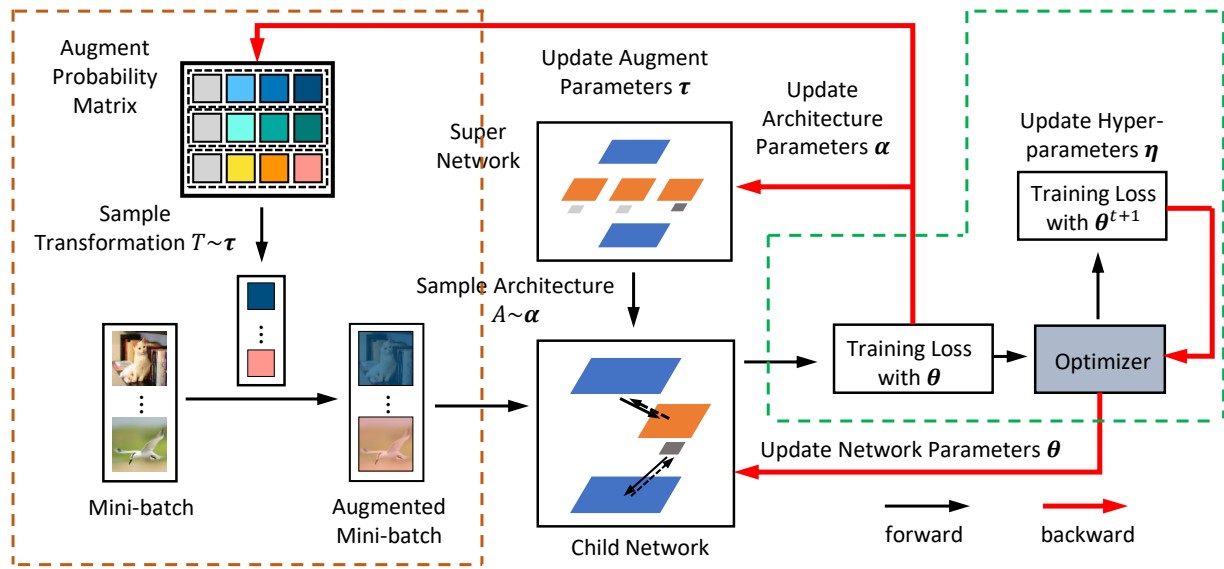

Figure 1: An overview of DHA. We first sample the DA operations for each sample based on the data transformation parameters $\boldsymbol{\tau}$. Then, a child network is sampled based on the architecture parameters $\boldsymbol{\alpha}$, which will be used to process the transformed mini-batch. Training loss is calculated to update the data transformation parameters $\boldsymbol{\tau}$, architecture parameters $\boldsymbol{\alpha}$, and neural network parameters $\boldsymbol{\theta}$. Then the training loss based on updated networks' weights $\boldsymbol{\theta}^{t+1}$ is used to update hyper-parameters $\boldsymbol{\eta}$.

## 2 Related Works

**Data augmentation.** Learning data augmentation policies for a target dataset automatically has become a trend, considering the difficulty to elaborately design augmentation methods for various datasets (Cubuk et al., 2018; Zhang et al., 2020; Lin et al., 2019). Specifically, AutoAugment (Cubuk et al., 2018) and Adversarial AutoAugment (Zhang et al., 2020) adopt reinforcement learning to train a controller to generate policies, while OHL-Auto-Aug (Lin et al., 2019) formulates augmentation policy as a probability distribution and adopts REINFORCE (Williams, 1992) to optimize the distribution parameters along with network training. PBA (Ho et al., 2019) and FAA (Lim et al., 2019) use population-based training method and Bayesian optimization respectively to reduce the computing cost of learning policies. Cubuk et al. (2020) argues that the search space of policies used by these works can be reduced greatly and simple grid search can achieve competitive performance. They also point out that the optimal data augmentation policy depends on the model size, which indicates that fixing an augmentation policy when searching for neural architectures may lead to sub-optimal solutions.

**Hyper-parameter optimization.** Hyper-parameters also play an important role in the training paradigm of deep neural network models. Various black-box optimization approaches have been developed to address hyper-parameter tuning tasks involving multiple tasks (Mittal et al., 2020; Perrone et al., 2018) or mixed variable types (Ru et al., 2020a). Meanwhile, techniques like multi-fidelity evaluations (Kandasamy et al., 2017; Wu et al., 2019), parallel computation (González et al., 2016; Kathuria et al., 2016; Alvi et al., 2019), and transfer learning (Swersky et al., 2013; Min et al., 2020) are also employed to further enhance the query efficiency of the hyper-parameter optimization. In addition, several works (Bengio, 2000; Lorraine et al., 2020; MacKay et al., 2019; Shaban et al., 2019; Maclaurin et al., 2015) have proposed to use gradient-based bi-level optimization to tune large number of hyper-parameters during model training. Although many HPO strategies have been adopted in NAS strategies, methods that jointly optimize both architectures and hyper-parameters are rarely seen except the ones discussed below.

**Neural architecture search.** NAS has attracted growing attention over the recent years and provided architectures with better performance over those designed by human experts (Pham et al., 2018; Real et al.,

2018; Liu et al., 2019). The rich collection of NAS literature can be divided into two categories: query-based methods and gradient-based ones. The former includes powerful optimization strategies such as reinforcement learning (Zoph & Le, 2017; Pham et al., 2018), Bayesian optimization (Kandasamy et al., 2018; Ru et al., 2020b) and evolutionary algorithms (Elsken et al., 2019; Lu et al., 2019). The latter enables the use of gradients in updating both architecture parameters and network weights, which significantly reducing the computation costs of NAS via weight sharing (Liu et al., 2019; Chen et al., 2019; Xie et al., 2019; Hu et al., 2020). To reduce searching cost, most NAS methods search architectures in a low-fidelity set-up (e.g.fewer training epochs, smaller architectures) and retrain the optimal architecture using the full set-up before deployment. This separation of *search* and *evaluation* is sub-optimal (Hu et al., 2020), which motivates the development of end-to-end NAS strategies (Xie et al., 2019; Hu et al., 2020) that return read-to-deploy networks at the end of the search. Our work also proposes an end-to-end solution.

**Joint optimization of AutoML components.** Conventional neural architecture search methods perform a search over a fixed set of architecture candidates and then adopt a separate set of hyper-parameters when retraining the best architecture derived from the architecture search phase. Such search protocol may lead to sub-optimal results (Zela et al., 2018; Dong et al., 2020) as it neglects the influence of training hyper-parameters on architecture performance and ignores superior architectures under alternative hyper-parameter values (Dai et al., 2020). Given this, several works have been proposed to jointly optimize architecture structure and training hyper-parameters (Dai et al., 2020; Wang et al., 2020; Dong et al., 2020). Zela et al. (2018) introduces the use of multi-fidelity Bayesian optimization to search over both the architecture structure and training hyper-parameters. Dai et al. (2020) trains an accuracy predictor to estimate the network performance based on both the architecture and training hyper-parameters and then uses an evolutionary algorithm to perform the search. Both these methods are query-based and require a relatively large number of architecture and hyper-parameter evaluations to fine-tune predictors or obtain good recommendations. To improve the joint optimization efficiency, AutoHAS (Dong et al., 2020) introduces a differentiable approach in conjunction with weight sharing for the joint optimization task, which empirically demonstrates that such a differentiable one-shot approach achieves superior efficiency over query-based methods. In addition to jointly optimizing neural architectures and hyper-parameters, the other line of research focuses on the joint optimization of neural architectures and data augmentation hyper-parameters (Kashima et al., 2020). Our proposed method differs from the above works in mainly two aspects: first, our method is more efficient than AutoHAS since our method has no need to update the whole network and only needs to update the sampled sub-network at each optimization step. Second, the joint optimization scope is further extended from NAS and training hyper-parameters to include DA, HPO and NAS.

## 3  Methodology

Consider a dataset $\mathcal{D} = \{(x_i, y_i)\}_{i=1}^{N}$, where $N$ is the size of this dataset, and $y_i$ is the label of the input sample $x_i$. We aim to train a neural network $f(\cdot)$, which can achieve the best accuracy on the test dataset $\mathcal{D}^{test}$. Multiple AutoML components are considered, including DA, HPO, and NAS. Let $\boldsymbol{\tau}$, $\boldsymbol{\eta}$, $\boldsymbol{\alpha}$, and $\boldsymbol{\theta}$ represent the data augmentation parameters, the hyper-parameters, the architecture parameters, and the objective neural network parameters, respectively. This problem can be formulated as

$$
\begin{aligned}
& argmin_{\boldsymbol{\tau},\boldsymbol{\eta},\boldsymbol{\alpha},\boldsymbol{\theta}} \mathcal{L}(\boldsymbol{\tau},\boldsymbol{\eta},\boldsymbol{\alpha},\boldsymbol{\theta};\mathcal{D}) \\
& s.t. \quad c_i(\boldsymbol{\alpha}) \leq C_i, i = 1,...,\gamma,
\end{aligned}
\tag{1}
$$

where $\mathcal{L}(\cdot)$ represents the loss function, $\mathcal{D}$ denotes the input data, $c_i(\cdot)$ refers to the resource cost (e.g., storage or computational cost) of the current architecture $\alpha$, which is restricted by the $i$-th resource constraints $C_i$, and $\gamma$ denotes the total number of resource constraints. Considering the huge search space, it is challenging to achieve the joint optimization of $\boldsymbol{\tau}$, $\boldsymbol{\eta}$, $\boldsymbol{\alpha}$, and $\boldsymbol{\theta}$ within one-stage without parameter retraining. In this work, we propose to use the differentiable method to provide a computationally efficient solution. See Fig. 1 for an illustration.

### 3.1 Data augmentation parameters

For every mini-batch of training data $\mathcal{B}^{tr} = \{(x_k, y_k)\}_{k=1}^{n^{tr}}$ with batch size $n^{tr}$, we conduct data augmentation to increase the diversity of the training data. We consider $K$ data augmentation operations, and each training sample is augmented by a transformation consisting of two successive operations (Cubuk et al., 2018; Lim et al., 2019). Each operation is associated with a magnitude that is uniformly sampled from $[0, 10]$. The data augmentation parameter $\boldsymbol{\tau}$ represents a probability distribution over the augmentation transformations. For $t$-th iteration, we sample $n^{tr}$ transformations according to $\boldsymbol{\tau}^t$ with Gumbel-Softmax reparameterization (Maddison et al., 2016) and to generate the corresponding augmented samples in the batch. Given a sampled architecture, the loss function for each augmented sample is denoted by $\mathcal{L}^{tr}(f(\boldsymbol{\alpha}^t, \boldsymbol{\theta}^t; \mathcal{T}_k(x_k)))$, where $\mathcal{T}_k$ represents the selected transformation. In order to relax $\boldsymbol{\tau}$ to be differentiable, we regard $p_k(\boldsymbol{\tau}^t)$, the probability of sampling the transformation $\mathcal{T}_k$, as an importance weight for the loss function of corresponding sample $\mathcal{L}^{tr}(f(\boldsymbol{\alpha}^t, \boldsymbol{\theta}^t; \mathcal{T}_k(x_k)))$. The objective of data augmentation is to minimize the following loss function:

$$\mathcal{L}^{DA}(\boldsymbol{\tau}^t) = -\sum_{k=1}^{n^{tr}} p_k(\boldsymbol{\tau}^t) \mathcal{L}^{tr}(f(\boldsymbol{\alpha}^t, \boldsymbol{\theta}^t; \mathcal{T}_k(x_k))). \tag{2}$$

With this loss function, DHA intends to increase the sampling probability of those transformations that can generate samples with high **training loss**. By sampling such transformations, DHA can pay more attention to more aggressive DA strategies and increase model robustness against difficult samples (Zhang et al., 2020). However, blindly increasing the difficulty of samples may cause the **augment ambiguity** phenomenon (Wei et al., 2020): augmented images may be far away from the majority of clean images, which could cause the under-fitting of model and deteriorate the learning process. Hence, besides optimizing the probability matrix of DA strategies, we randomly sample the magnitude of each chosen strategy from an **uniform distribution**, which can prevent learning heavy DA strategies: augmenting samples with large magnitude strategies. Moreover, instead of training a controller to generate adversarial augmentation policies via reinforcement learning (Zhang et al., 2020) or training an extra teacher model to generate additional labels for augmented samples (Wei et al., 2020), we search for the probability distribution of augmentation transformations directly via gradient-based optimization. In this way, the optimization of data augmentation is very efficient and hardly increases the computing cost.

### 3.2 Hyper-parameters

As shown in Fig. 1, given the batch of augmented training data $\{(\mathcal{T}_k(x_k), y_k)\}_{k=1}^{n^{tr}}$ and the sampled child network, we need to optimize the differentiable hyper-parameters $\boldsymbol{\eta}$, such as learning rate and L2 regularization. At the training stage, we alternatively update $\boldsymbol{\theta}$ and $\boldsymbol{\eta}$. In $t$-th iteration, we can update $\boldsymbol{\theta}^t$ based on the gradient of the unweighted training loss $\mathcal{L}^{tr}(f(\boldsymbol{\alpha}^t, \boldsymbol{\theta}^t; \mathcal{B}^{tr})) = \frac{1}{n^{tr}} \sum_{k=1}^{n^{tr}} \mathcal{L}^{tr}(f(\boldsymbol{\alpha}^t, \boldsymbol{\theta}^t; \mathcal{T}_k(x_k)))$, which can be written as:

$$\boldsymbol{\theta}^{t+1} = OP(\boldsymbol{\theta}^t, \boldsymbol{\eta}^t, \nabla_{\boldsymbol{\theta}} \mathcal{L}^{tr}(f(\boldsymbol{\alpha}^t, \boldsymbol{\theta}^t; \mathcal{B}^{tr}))), \tag{3}$$

where $OP(\cdot)$ is the optimizer. To update the hyper-parameters $\boldsymbol{\eta}$, we regard $\boldsymbol{\theta}^{t+1}$ as a function of $\boldsymbol{\eta}$ and compute the **training loss** $\mathcal{L}^{tr}(f(\boldsymbol{\alpha}^t, \boldsymbol{\theta}^{t+1}(\boldsymbol{\eta}^t); \mathcal{B}^{tr}))$ with network parameters $\boldsymbol{\theta}^{t+1}(\boldsymbol{\eta}^t)$ on a mini-batch of training data $\mathcal{B}^{tr}$. Then, $\boldsymbol{\eta}^t$ is updated with $\nabla_{\boldsymbol{\eta}} \mathcal{L}^{tr}(f(\boldsymbol{\alpha}^t, \boldsymbol{\theta}^{t+1}(\boldsymbol{\eta}^t); \mathcal{B}^{tr}))$ by gradient descent:

$$\boldsymbol{\eta}^{t+1} = \boldsymbol{\eta}^t - \beta \nabla_{\boldsymbol{\eta}} \mathcal{L}^{tr}(f(\boldsymbol{\alpha}^t, \boldsymbol{\theta}^{t+1}(\boldsymbol{\eta}^t); \mathcal{B}^{tr})), \tag{4}$$

where $\beta$ is a learning rate. Even $\boldsymbol{\theta}^t$ can also be deployed to $\boldsymbol{\theta}^{t-1}$ whose calculation also involves $\boldsymbol{\eta}^{t-1}$, we take an approximation method in Eqn. (4) and regard $\boldsymbol{\theta}^t$ here as a variable independent of $\boldsymbol{\eta}^{t-1}$. Instead of splitting an extra validation set for HPO, we directly sample a subset from training set to update $\boldsymbol{\eta}$, which could ensure that the whole training set is used in updating $\boldsymbol{\tau}$, $\boldsymbol{\eta}$, $\boldsymbol{\alpha}$, and $\boldsymbol{\theta}$. This avoids the final learned weight decay coefficient to be zero as non-zero weight decay coefficient can help avoid the model to overfit to the training data.

### 3.3 Architecture parameters

With the augmented data in previous section, we achieve the optimization of the architecture parameter $\boldsymbol{\alpha}$ through end-to-end NAS, motivated by SNAS (Xie et al., 2019), DSNAS (Hu et al., 2020) and ISTA-

---

**Algorithm 1** DHA

---

**Initialization:** Data Transformation Parameters $\boldsymbol{\tau}$, Hyper-parameters $\boldsymbol{\eta}$, Compressed Representation $\mathbf{b}$, Measurement Matrix $\mathbf{A}$, and Network Parameters $\boldsymbol{\theta}$
**Input:** Training Set $\mathcal{D}^{tr}$, Parameters $\boldsymbol{\tau}$, $\boldsymbol{\eta}$, $\mathbf{b}$, $\mathbf{A}$, $\boldsymbol{\theta}$, and the iteration number $T$
**Return:** $\boldsymbol{\tau}, \boldsymbol{\eta}, \boldsymbol{\alpha}, \boldsymbol{\theta}$

1: **while** $t < T$ **do**
2:     Separately sample a mini-batch $\mathcal{B}^{tr}$ from $\mathcal{D}^{tr}$;
3:     For each sample $x_k$ in mini-batch $\mathcal{B}^{tr}$, sample a transformation $\mathcal{T}_k(x_k)$ according to $\boldsymbol{\tau}^t$;
4:     Recover $\boldsymbol{\alpha}^t$ by solving Eqn. (5) with $\mathbf{b}^t$ and $\mathbf{A}$;
5:     Extract a child network from the super network;
6:     Compute the weighted training loss function as Eqn. (2) and update $\boldsymbol{\tau}^{t+1}$ accordingly;
7:     Calculate $\boldsymbol{\theta}^{t+1}$ with Eqn. (3);
8:     Use training loss function to update $\mathbf{b}^{t+1}$ through the gradient descent, then $\boldsymbol{\alpha}^{t+1}$ is updated with Eqn. (6);
9:     Compute the training loss function with $\boldsymbol{\theta}^{t+1}$ on $\mathcal{D}^{val}$ and update $\boldsymbol{\eta}^{t+1}$ with Eqn. (4);
10: **end while**

---

NAS (Yang et al., 2020). Following Liu et al. (2019), we denote the each space as a single directed acyclic graph (DAG), where the probability matrix $\boldsymbol{\alpha}$ consists of vector $\boldsymbol{\alpha}_{i,j}^T = [\alpha_{i,j}^1, ..., \alpha_{i,j}^r, ..., \alpha_{i,j}^k]$ and $\alpha_{i,j}^r$ represents the probability of choosing $r^{th}$ operation associated with the edge $(i, j)$. Instead of directly optimizing $\boldsymbol{\alpha} \in \mathbb{R}^n$, we adopt ISTA-NAS to optimize its compressed representation $\mathbf{b} \in \mathbb{R}^m$ where $m << n$, which can be written as:

$$\mathbf{b} = \mathbf{A}\boldsymbol{\alpha} + \epsilon, \tag{5}$$

where $\epsilon \in \mathbb{R}^m$ represents the noise and $\mathbf{A} \in \mathbb{R}^{m \times n}$ is the measurement matrix which is randomly initialized. Eqn. (5) is solved through using LASSO loss function (Tibshirani, 1996) and the $\boldsymbol{\alpha}$ is optimized by using iterative shrinkage thresholding algorithm (Daubechies et al., 2004), which can be written as:

$$\boldsymbol{\alpha}^{t+1} = \eta_{\lambda/L}(\boldsymbol{\alpha}^t - \frac{1}{L}\mathbf{A}^T(\mathbf{A}\boldsymbol{\alpha} - \mathbf{b})), t = 0, 1, ..., \tag{6}$$

where $L$ represents the LASSO formulation which can be written as $\min_{\boldsymbol{\alpha}} \frac{1}{2}||\mathbf{A}\boldsymbol{\alpha} - \mathbf{b}||_2^2 + \lambda||\boldsymbol{\alpha}||_1$; the $\lambda$ represents the regularization parameters and the $\eta_{\lambda/L}$ is the shrinkage operator as defined in (Beck & Teboulle, 2009). Thus we have:

$$\boldsymbol{\alpha}_j^T \mathbf{o}_j = (\mathbf{b}_j^T \mathbf{A}_j - [\boldsymbol{\alpha}_j(\mathbf{b}_j)]^T \mathbf{E}_j)\mathbf{o}_j, \tag{7}$$

where $\mathbf{o}_j$ refers to all possible operations connected to note $j$ and $\mathbf{E}_j = \mathbf{A}_j^T \mathbf{A}_j - \mathbf{I}$. With this relaxation, $\mathbf{b}$ can be optimized through calculating the gradient concerning **training loss**. The main reason for using this optimization algorithm is that it can optimize the high-dimensional architecture parameters through optimizing low-dimensional embeddings, which can largely decrease the optimization difficulty and increase the optimization efficiency. Moreover, this algorithm also adopt the weights sharing in the optimization process which can be readily combined with our proposed data augmentation and hyper-parameter optimization method.

### 3.4   Joint-optimization

Based on the above analysis of each AutoML module, DHA realizes end-to-end joint optimization of automated data augmentation parameters $\boldsymbol{\tau}$, hyper-parameters $\boldsymbol{\eta}$, and architecture parameters $\boldsymbol{\alpha}$. The DHA algorithm is summarized in Algorithm 1. One-level optimization is applied to $\boldsymbol{\tau}$ and $\boldsymbol{\alpha}$ as in Line 6 and Line 8, while bi-level optimization is applied to $\boldsymbol{\eta}$ as in Line 9.

One thing worth mentioning is that different optimizers are adopted for different parameters. There are two main reasons behind this choice. Firstly, $\boldsymbol{\tau}, \boldsymbol{\eta}, \boldsymbol{\alpha}$ and $\boldsymbol{\theta}$ work differently in the optimization process, e.g., $\boldsymbol{\tau}$ controls the transformation strategy for the training set while $\boldsymbol{\alpha}$ is related to the architecture selection. Besides $\boldsymbol{\theta}$, other parameters could not directly be optimized through the gradient descent on training set. These differences cause the different optimization methods for $\boldsymbol{\tau}, \boldsymbol{\alpha}, \boldsymbol{\eta}$ and $\boldsymbol{\theta}$. Secondly, $\boldsymbol{\tau}, \boldsymbol{\alpha}, \boldsymbol{\eta}$ and $\boldsymbol{\theta}$ have

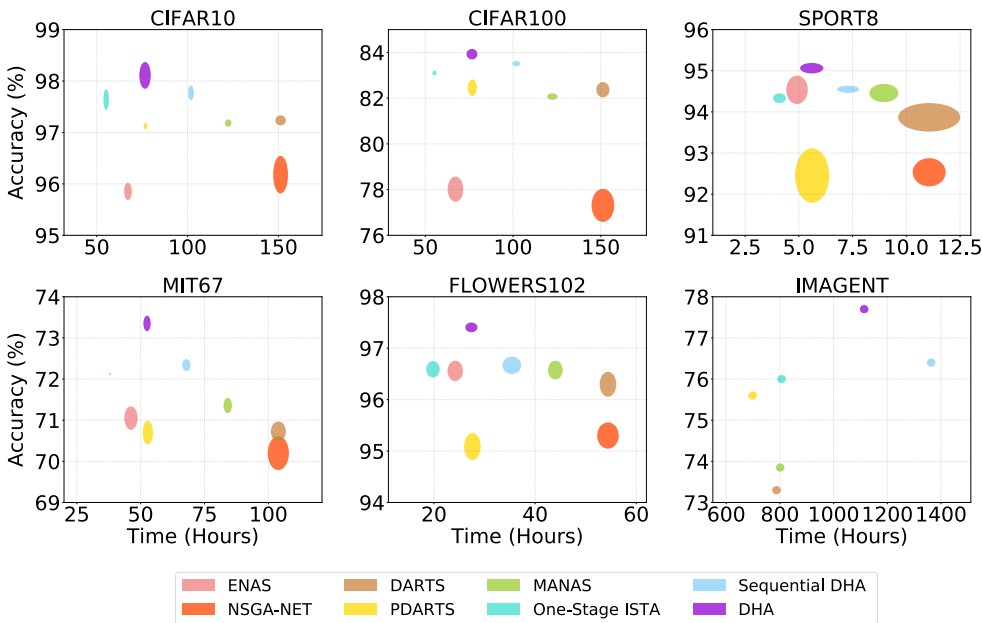

Figure 2: Top-1 accuracy and computational time of AutoML algorithms for classification task on CIFAR10, CIFAR100, SPORT8, MIT67, FLOWERS102 and ImageNet. DHA is conducted with Cell-Based search space. Ellipse centres and ellipse edges represent the $\mu \pm \{0, \sigma/2\}$, respectively (mean $\mu$, standard deviation $\sigma$). For ImageNet, we present test accuracy without error bars, as the error bars are not reported in existing works.

different dimensions and different scales, which makes the joint-optimization with a uniform optimizer extremely impracticable. This is why current works concerning DA (Ho et al., 2019), NAS (Yao et al., 2020; Nekrasov et al., 2019) and HPO (Falkner et al., 2018) always adopt different optimizer for network's weights and hyper-parameters they aim to optimize. During the training process, DA and HPO adapt the online optimization strategy. The DA strategy and HPO settings are evolving with the weights of the parameters.

Moreover, the main reason that DHA could optimize large-scale search space in an effectively manner, is that DHA delicately adopt **weight-sharing** in the joint-optimization for different parameters. Instead of only optimizing a sub-network with a DA strategy and a hyper-parameter setting to check the performance of certain setting, we realize the joint-optimization with the help of a super-net network, a DA probability matrix and continuous hyper-parameter setting. In that way, DHA can make use of previous trained parameter weights to check the performance setting, which largely decreases the computational request.

## 4 Experiments

In this section, we empirically compare DHA against existing AutoML algorithms on various datasets. With extensive experiments, we demonstrate the benefits of joint-optimization over sequential-optimization in terms of generalization performance and computational efficiency.

### 4.1 Experiment setting

**Datasets.** Following Ru et al. (2020b), we conducted experiments on various datasets, including CIFAR10 and CIFAR100 for the object classification task (Krizhevsky et al., 2009), SPORT8 for the action classification task (Li & Fei-Fei, 2007), MIT67 for the scene classification task (Quattoni & Torralba, 2009), FLOWERS102 for the fine object classification task (Nilsback & Zisserman, 2008), and ImageNet(Russakovsky et al., 2015) for the large-scale classification task. The accuracy is calculated on the test set.

**Search space. (1) Automated DA.** Following Ho et al. (2019), we consider $K = 14$ different operations for data augmentation, such as AutoContrast and Equalize. The magnitude of each operation is randomly sampled from the uniform distribution. Note that we follow a widely used data augmentation setting (Zhang et al., 2020; Zhou et al., 2021), which applies two successive data augmentation strategies on each training

Table 1: Top-1 accuracy (%) and computational time (GPU hour) of different AutoML algorithms on CIFAR10, CIFAR100, SPORT8, MIT67 and FLOWERS102 with Cell-Based Search Space. We report the sum of search time and tune time for two-stage NAS and the whole running time for one-stage NAS. All the methods are implemented by ourselves. Different NAS algorithms for one dataset are performed under the similar parameter weights constrain to ensure fair comparison.

| Model | CIFAR10 Acc | CIFAR100 Acc | SPORT8 Acc | MIT67 Acc | FLOWERS102 Acc |
|---|---|---|---|---|---|
| ENAS (Pham et al., 2018) | 95.85±0.17 | 78.02±0.55 | 94.54±0.35 | 71.05±0.29 | 96.56±0.20 |
| NSGA-NET (Lu et al., 2019) | 96.18±0.37 | 77.31±0.14 | 92.53±0.34 | 70.20±0.41 | 95.30±0.26 |
| DARTS (Liu et al., 2019) | 97.24±0.10 | 82.37±0.34 | 93.87±0.35 | 70.73±0.24 | 96.30±0.24 |
| P-DARTS (Chen et al., 2019) | 97.13±0.07 | 82.46±0.37 | 92.45±0.66 | 70.70±0.29 | 95.09±0.26 |
| MANAS (Carlucci et al., 2019) | 97.18±0.07 | 82.07±0.14 | 94.46±0.22 | 71.36±0.19 | 96.57±0.18 |
| One-Stage ISTA (Yang et al., 2020) | 97.64±0.20 | 83.10±0.11 | 94.33±0.12 | 72.12±0.03 | 96.59±0.16 |
| Sequential DHA | 97.77±0.14 | 83.51±0.12 | 94.55±0.09 | 72.34±0.14 | 96.67±0.17 |
| DHA | **98.11**±0.26 | **83.93**±0.23 | **95.06**±0.13 | **73.35**±0.19 | **97.41**±0.09 |
| | **Time** | **Time** | **Time** | **Time** | **Time** |
| ENAS (Pham et al., 2018) | 67.2±2.1 | 67.2±4.4 | 4.92±0.5 | 46.2±2.6 | 24.2±1.5 |
| NSGA-NET (Lu et al., 2019) | 151.2±4.1 | 151.2±6.5 | 11.06±0.8 | 103.9±4.1 | 54.4±2.1 |
| DARTS (Liu et al., 2019) | 151.2±2.9 | 151.2±3.8 | 11.06±1.4 | 103.9±3 | 54.4±1.6 |
| P-DARTS (Chen et al., 2019) | 76.8±1 | 76.8±2.6 | 5.62±0.8 | 52.8±2 | 27.6±1.6 |
| MANAS (Carlucci et al., 2019) | 122.4±1.8 | 122.4±2.9 | 8.96±0.7 | 84.1±1.6 | 44±1.5 |
| One-Stage ISTA (Yang et al., 2020) | 55.2±1.5 | 55.2±1.2 | 4.1±0.3 | 37.9±0.4 | 19.8±1.3 |
| Sequential DHA | 101.9±1.6 | 101.9±2.3 | 7.3±0.5 | 67.9±1.6 | 35.4±1.9 |
| DHA | 76.6±3.2 | 76.6±3.1 | 5.6±0.5 | 52.5±1.4 | 27.4±1.2 |

sample. Our method can be easily extended to more general cases by modifying the probability matrix and the range of sampling distribution. For example, for the case of three successive data augmentation strategies, we could extend the size of probability matrix from $K^2$ to $K^3$, where $K$ is the number of data augmentation categories. **(2) NAS.** Following Liu et al. (2019) and Cai et al. (2019), we consider both the cell-based and the MobileNet search space, which regards the whole architecture as a stack similar cells. **(3) HPO.** We consider both the L2 regularization (i.e., weight decay) and the learning rate in the experiments. Detailed information is provided in Appendix A.

**Baselines.** We compare DHA with various AutoML algorithms, such as ENAS (Pham et al., 2018), NSGA-NET (Lu et al., 2019) , NSGA-NET (Lu et al., 2019) , P-DARTS (Chen et al., 2019), MANAS (Carlucci et al., 2019) and One-Stage ISTA (Yang et al., 2020)(see Table 1 and Table 2). For datasets including CIFAR10, CIFAR100, SPORT8, MIT67, and FLOWERS102, we re-implement the baseline methods. Specifically, for papers with code, incling ENAS, NSGA-NET, P-DARTS and One-Stage ISTA, we directly use their official implementation code. For paper without code, including DARTS and MANAS, we use the unofficial re-implementation code which achieves similar performance described in the original papers. As for ImageNet, we directly refer to the model performance reported in papers cited in Table 2. To further demonstrate the benefits of joint optimization of multiple AutoML components, we also include a baseline, **Sequential DHA**, which resembles the common practice by human to optimize different components in sequence. Specifically, Sequential DHA consists of two stages. During the first stage, Sequential DHA performs NAS to find the optimal architecture under certain hyper-parameter settings. In the next stage, Sequential DHA performs the online DA and HPO strategy proposed in our paper and trains the architecture derived from the first stage from scratch. Detailed algorithm of Sequential DHA could be found in Appendix B.

**Implementation details.** For hyper-parameters of DHA, we simply adopt the hyper-parameters used in previous works without many modifications, including Meta-Aug (Zhou et al., 2021), ISTA-NAS (Yang et al., 2020), and DSNAS (Hu et al., 2020). Experiments are conducted on NVIDIA V100 under PyTorch-1.3.0 and Python 3.6. Detailed settings of baselines are provided in Appendix A.

Table 2: Comparison with SOTA image classifiers on ImageNet in the Cell-Based (C) setting or MobileNet/ShuffleNet (M) setting. ($\dagger$ denotes the architecture is searched on ImageNet, otherwise it is searched on CIFAR-10 or CIFAR-100.)

| Model | Test Acc (%) | | Params (M) | Cost (GPU-day) | | Search Attribute | Search Space |
|---|---|---|---|---|---|---|---|
| | Top-1 | Top-5 | | Search | Eval | | |
| DARTS (2nd) (Liu et al., 2019) | 73.3 | 91.3 | 4.7 | 4.0 | $3.6 \times 8$ | Arch | C |
| SNAS (mild) (Xie et al., 2019) | 72.7 | 90.8 | 4.3 | 1.5 | $3.3 \times 8$ | Arch | C |
| GDAS (Dong & Yang, 2019) | 74.0 | 91.5 | 5.3 | 0.3 | $3.6 \times 8$ | Arch | C |
| BayesNAS (Zhou et al., 2019) | 73.5 | 91.1 | 3.9 | 0.2 | $3.6 \times 8$ | Arch | C |
| PARSEC (Casale et al., 2019) | 74.0 | 91.6 | 5.6 | 1.0 | $3.6 \times 8$ | Arch | C |
| P-DARTS (CIFAR-10) (Chen et al., 2019) | 75.6 | 92.6 | 4.9 | 0.3 | $3.6 \times 8$ | Arch | C |
| P-DARTS (CIFAR-100) (Chen et al., 2019) | 75.3 | 92.5 | 5.1 | 0.3 | $3.6 \times 8$ | Arch | C |
| PC-DARTS (ImageNet) (Xu et al., 2020)$^\dagger$ | 75.8 | 92.7 | 5.3 | 3.8 | $3.9 \times 8$ | Arch | C |
| GAEA+PC-DARTS (Li et al., 2021)$^\dagger$ | 76.0 | 92.7 | 5.6 | 3.8 | $3.9 \times 8$ | Arch | C |
| DrNAS (Chen et al., 2021)$^\dagger$ | 76.3 | 92.9 | 5.7 | 4.6 | $3.9 \times 8$ | Arch | C |
| Two-Stage ISTA (Yang et al., 2020)$^\dagger$ | 75.0 | 91.9 | 5.3 | 2.3 | $3.4 \times 8$ | Arch | C |
| One-Stage ISTA (Yang et al., 2020)$^\dagger$ | 76.0 | 92.9 | 5.7 | | $4.2 \times 8$ | Arch | C |
| EfficientNet-B0 (Tan & Le, 2019)$^\dagger$ | 77.1 | 93.3 | 5.3 | - | - | Arch | M |
| SinglePathNAS (Guo et al., 2019)$^\dagger$ | 74.7 | - | 3.4 | 13.0 | $2.0 \times 8$ | Arch | M |
| ProxylessNAS (GPU) (Cai et al., 2019)$^\dagger$ | 75.1 | 92.5 | 7.1 | 8.3 | $3.6 \times 8$ | Arch | M |
| DSNAS (Hu et al., 2020)$^\dagger$ | 74.3 | 91.9 | - | | $3.7 \times 8$ | Arch | M |
| OFA (small) (Cai et al., 2020)$^\dagger$ | 76.9 | 93.3 | 5.8 | | $6.8 \times 8$ | Arch + Resolution | M |
| APQ (Wang et al., 2020)$^\dagger$ | 75.1 | - | - | | $12.5 \times 8$ | Arch + Pruning | M |
| AutoHAS (Dong et al., 2020) $^\dagger$ | 74.2 | - | - | - | - | Arch + HPO | M |
| Sequential DHA$^\dagger$ | 76.7 | 93.8 | 5.4 | 3.2 | $5.5 \times 8$ | Arch + DA + HPO | C |
| DHA$^\dagger$ | 77.4 | 94.6 | 5.6 | | $5.8 \times 8$ | Arch + DA + HPO | C |
| Sequential DHA$^\dagger$ | 77.1 | 93.9 | 5.7 | 5.8 | $5.7 \times 8$ | Arch + DA + HPO | M |
| DHA$^\dagger$ | **77.6** | 94.8 | 5.3 | | $5.4 \times 8$ | Arch + DA + HPO | M |

Table 3: Top-1 accuracy (%) and computational time (GPU hour) of different combination of AutoML components on CIFAR10, CIFAR100, SPORT8, MIT67 and FLOWERS102. Listed algorithms are described in Ablation study.

| Model | CIFAR10 | | CIFAR100 | | SPORT8 | | MIT67 | | FLOWERS102 | |
|---|---|---|---|---|---|---|---|---|---|---|
| | Acc | Time | Acc | Time | Acc | Time | Acc | Time | Acc | Time |
| Sequential NAS+DA | 97.74 | 57.50 | 83.45 | 57.50 | 94.49 | 4.20 | 72.22 | 39.20 | 96.64 | 20.50 |
| Sequential NAS+HPO | 97.54 | 63.60 | 83.13 | 63.60 | 94.47 | 4.60 | 72.10 | 43.10 | 96.57 | 22.50 |
| Sequential DHA | 97.77 | 101.90 | 83.51 | 101.90 | 94.55 | 7.30 | 72.34 | 67.90 | 96.67 | 35.40 |
| Joint-optimization NAS+DA | 97.78 | 64.00 | 83.55 | 64.00 | 94.53 | 4.70 | 72.38 | 43.80 | 96.72 | 22.90 |
| Joint-optimization NAS+HPO | 97.75 | 70.80 | 83.23 | 70.80 | 94.50 | 5.20 | 72.23 | 47.80 | 96.69 | 25.40 |
| DHA | **98.11** | 76.60 | **83.93** | 76.60 | **95.06** | 5.60 | **73.35** | 52.50 | **97.41** | 27.40 |

## 4.2 Results

The test accuracy and computational time of various AutoML algorithms are summarized in Table 1 and Table 2. The timing results in these two tables measure the computational time taken to obtain a ready-to-deploy network, which corresponds to the sum of search and retrain time for two-stage NAS methods and the search time for end-to-end methods like one-stage NAS method as well as our DHA.

**Small-scale datasets.** As shown in Table 1, methods optimizing all of DA, HPO and NAS automatically (i.e, Sequential DHA and DHA) consistently outperform those NAS algorithms with manual designed DA and HPO. Specifically, DHA achieves SOTA results on all datasets. This shows the clear performance gain of extending the search scope from architecture to including also data augmentation and hyper-parameters, justifying the need for multi-component optimization in AutoML. Moreover, despite optimising over a larger search space, DHA remains cost efficiency. For example, on CIFAR100, DHA enjoys 1.56% higher test accuracy than DARTS but requires 42% less time. Besides, the comparison between DHA and Sequential DHA reveals the evident advantage of doing DA, HPO and NAS jointly over doing them separately in different stages.

**Large-scale dataset.** Results of the large-scale dataset ImageNet with cell-based search space and MobileNet search space are shown in Table 2. DHA consistently outperforms various NAS methods which only involves

architecture optimization, demonstrating the benefits of joint-optimization. Even when compared with One-stage NAS methods like ISTA, DHA achieves up to 1.7% TOP-1 accuracy improvement. Moreover, in comparison with the joint-optimization algorithm APQ (Wang et al., 2020) and AutoHAS (Dong et al., 2020), DHA outperforms APQ by 2.5% and outperforms AutoHAS by 4.3%. These comparisons reveal that DHA proposes an efficient and high-performed joint-optimization algorithm. The Top-1 accuracy and computation time of these AutoML algorithms are also summarized in Fig. 2. As can be seen, DHA consistently gains highest test accuracy on all five datasets while being more cost efficient than NAS methods and Sequential DHA. The validation accuracy of Sequential-DHA and DHA during the training is shown in Fig. 3. We could notice that a traditional sequential optimization algorithm would disrupt the learning process, as it has to retrain the model after the searching phase. While benefiting from the end-to-end optimization process, DHA would have a smoother learning curve and also achieve better performance.

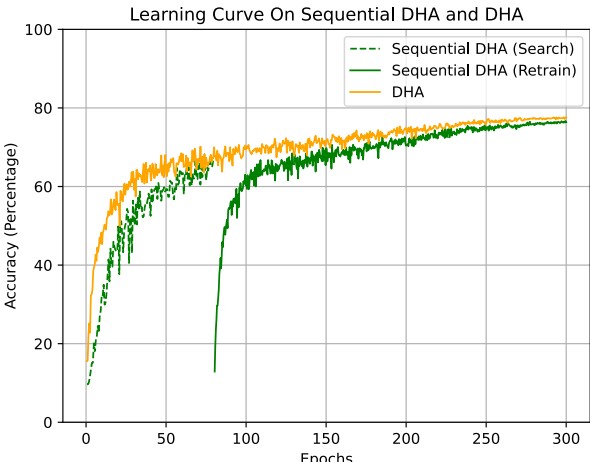

Figure 3: The validation accuracy of Sequential-DHA and DHA over the training time on ImageNet with Cell-Based Structure.

## 4.3  Analysis of loss landscape

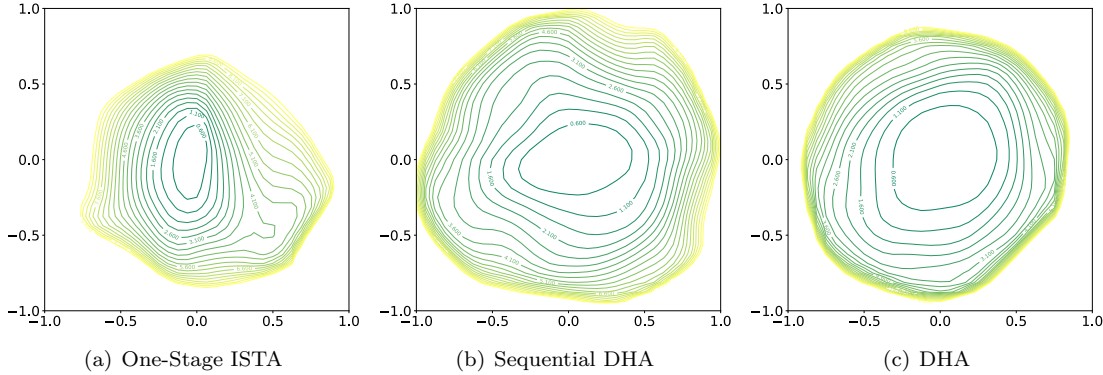

(a) One-Stage ISTA          (b) Sequential DHA          (c) DHA

Figure 4: Loss-landscape of the trained models on SPORT8. (a) One-Stage ISTA (b) Sequential DHA and (c) DHA. The strong test performance of DHA across various datasets motivates us to further check the geometry of the minimiser achieved by our final trained model. We employ the filter-normalisation method proposed in Li et al. (2017) to visualise the local loss landscape around the minimiser achieved by One-Stage ISTA, Sequential DHA and DHA. We choose a center point $\boldsymbol{\theta}^*$ in the graph, and two direction vector $\boldsymbol{\delta}$ and $\boldsymbol{\eta}$. We the plot the 2D contour with the function $f(\alpha, \beta) = \mathcal{L}(f(\boldsymbol{\theta}^* + \alpha\boldsymbol{\delta} + \beta\boldsymbol{\eta}; \mathcal{B}^{test}))$, where $\mathcal{B}^{test}$ represents the test set. The resultant contour plots are shown in Fig. 4. As can be seen, the minimum reached by optimising over more AutoML components tend to be flatter than NAS with manual designed DA and HPO.

The loss landscape of DHA is also flatter and smoother than that of Sequential DHA accounting for the better generalisation performance of DHA in previous experiments, explaining the superior test accuracy achieved by DHA in Tables 1 and 2 (Keskar et al., 2016; Xu & Mannor, 2012).

### 4.4 Ablation study of AutoML components

To further verify the effectiveness of our proposed extended search scope and search strategy, we empirically investigate the performance of considering all three components (i.e., DA, HPO and NAS) against combining any two of them. We examine both the sequential-optimization and joint-optimization settings with four designed optimization algorithms. **Sequential NAS + DA** first conducts NAS and during the tuning stage, the proposed DA optimization is applied. Similar to the Sequential NAS+DA, **Sequential NAS + HPO** firstly conducts NAS. Then with the fixed architecture, Sequential NAS + DA optimizes the HPO and networks' weights simultaneously. In contrast to them, **Joint-optimization NAS + DA** simultaneously conducts DA strategy optimization, NAS and parameter weights optimization. **Joint-optimization NAS + HPO** simultaneously conducts HPO, NAS and parameter weights optimization.

The comparison results are presented in Table 3. We can notice that performing joint optimization for either NAS + DA or NAS + HPO, achieves higher test accuracy than doing them in a pipeline. This reconfirms our previous conclusion that optimising different AutoML components jointly is better than doing them in sequence. Moreover, by comparing the results of the joint-optimization NAS+DA and the joint-optimization NAS+HPO in Table 3 against the One-Stage ISTA in Table 1, it is clear that considering one more AutoML component on top of NAS can lead to clear performance gain of the final model. While such gain is higher for incorporating DA than HPO, it is maximised when all three components are considered; our DHA obtains the best test accuracy among all joint-optimization baselines across all datasets.

We also provide (1) more ablation studies of DA and HPO (see Appendix C), (2) analysis of the searched architectures (see Appendix D) and comparison in terms of FLOPs and Latency (see Appendix E). Please refer to the Appendix for more details.

## 5 Conclusion

In this work, we present DHA, an end-to-end joint-optimization method for three important components of AutoML, including DA, HPO and NAS. This differentiable joint-optimization method can efficiently optimize larger search space than precious AutoML methods and achieve SOTA results on various datasets with a relatively low computational cost. Specifically, DHA achieves 77.4% Top-1 accuracy on ImageNet with cell based search space, which is higher that current SOTA by 0.5%. With DHA, we show the advantage of doing joint-optimization of AutoML over doing co-optimization in sequence, and conclude that joint optimization of multiple AutoML components is necessary.

## 6 Acknowledgments

We gratefully acknowledge the support of MindSpore, CANN (Compute Architecture for Neural Networks) and Ascend AI Processor used for this research.

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
