# OpenReview forum: "DHA: End-to-End Joint Optimization of Data Augmentation Policy, Hyper-parameter and Architecture"
_TMLR — Accepted by TMLR_

### Review · Reviewer_HSAw · 2022-08-02

**Summary Of Contributions:**

In this paper, the authors proposed DHA, an end-to-end AutoML method to optimize data augmentation (DS), hyper-parameters (HPO), and model architectures (NAS) in the same loop. By using weight-sharing for super network, data augmentatio0n probability matrix, and continuous hyper-parameter setting, DHA can efficiently explore the large search space. Experiments on multiple datasets show that DHA outperforms existing work and sequential optimization methods.

**Broader Impact Concerns:**

I don't see there is a prominent ethical concern in this paper.

**Requested Changes:**

As discussed above, it would be better to:
- Compare with existing AutoML work on their specific optimization setting (e.g., FLOPs, latency, etc.) instead of just comparing the model size.
- Discuss the difficulty of hyper-parameter tuning of the proposed method. Do we need to carefully tune the experimental setting to get good results?

**Strengths And Weaknesses:**

Pros:
- Firstly, the paper is generally well written and easy to follow.
- Solid ablation studies in Table 3 on smaller datasets show the superior performance of joint optimization.
- The loss landscape visualization results confirm that the proposed method leads to better optimization.
- It targets a very ambitious target to perform end-to-end optimization on the whole training procedure.

Cons:
- There is only parameter-constrained search, which may not provide a fair comparison to existing work since many of them are designed for computation efficiency (e.g., EfficientNet) or hardware efficiency/latency (e.g., ProxylessNAS).
- For hyper-parameter optimization, the design space is limited to weight decay and learning rate. Is there any possibility of extending to more hyper-parameter settings like optimizers (SGD vs. Adam)?
- The proposed method optimizes three components in the same loop. Can the author discuss the difficulty of hyper-parameter adjustment (in the DHA method itself but not training)? Is the adjustment easy and general? I am asking because some design knobs seem non-trivial (e.g., use different optimizers for different parameters).

---

### Review · Reviewer_JcoF · 2022-08-08

**Summary Of Contributions:**

This paper studies end-to-end optimization of data augmentation policy, hyper-parameter, and neural architecture. Existing pipelined solutions are time-consuming and sub-optimal. Therefore, the authors introduce DHA to offer an end-to-end solution. Technically, the authors formulate data augmentation, hyper-parameter, and neural architecture in continuous spaces to be optimized differentiably. The authors have evaluated their proposed DHA on multiple small-scale benchmarks and on ImageNet, achieving encouraging results.

**Requested Changes:**

Please refer to the weaknesses section for more details.

**Strengths And Weaknesses:**

---
**Strengths:**

The problem studied in this paper is essential in real-world scenarios. Finding the proper data augmentation, hyper-parameter, and neural architecture is very challenging in practice. Therefore, it is vital to have a push-the-button solution that could achieve good performance. Though simple, the proposed DHA could be a valuable first step towards this goal.

The paper is well-written and easy to follow. The authors formulate the problem at the beginning of Section 3 and provide sufficient technical details in the following subsections. The authors also include some discussions at the end of Section 3 to help the readers better understand the relationship and differences between previous solutions and the proposed DHA.

---
**Weaknesses:**

The authors have only provided the accuracy and model size in Table 2. This is not very informative as most baselines are searched under either computational cost (#FLOPs) or latency constraints. It would be necessary to report these two efficiency metrics for an apples-to-apples comparison with the baselines. As the results on small-scale datasets are not representative, the authors could consider evaluating their proposed DHA on other large-scale benchmarks (such as COCO object detection). Even a direct policy transfer would be sufficient.

From Table 2, the improvement of joint optimization is not very significant over sequential optimization (i.e., 0.5-0.7% top-1 accuracy). This makes it a bit questionable whether it is necessary to perform the joint optimization (especially when considering the extra engineering efforts needed). It would be great if the author could share some insights and thoughts on this.

The differentiable formulation is very nice since it can be effectively optimized using gradient-based methods. However, it is unclear whether it can cover all sorts of data augmentation policies and hyper-parameters. For instance, adding the quantization into the search space seems a bit challenging with the current differentiable formulation. The authors could comment on this.

The proposed pipeline requires a considerable amount of engineering effort to implement. Therefore, it would be super helpful (but no pressure!) if the authors could release their code upon the paper's acceptance to facilitate the reproduction.

---

### Review · Reviewer_HZRV · 2022-08-23

**Summary Of Contributions:**

The paper explores joint Data Augmentation, HPO and NAS. As the authors claim, it is rare to see joint DHA optimization due to its expensive nature. They key to implemnting joint optimization is to create a differential space parameterized individually for 'D', 'H' and 'A', followed by "One-level optimization is applied to 'D' parameter and the 'H' parameter, while bi-level optimization is applied to 'A' parameter."

**Requested Changes:**

- [Minor] Can Figure 1 be moved to an appropriate location (are these results?)
- [Minor] Without any context in page 2, "weight sharing is delicately applied.... super-network" does not make sense. All these all terms in Xie et al? What is "delicately adopting"?
-  Page 2 - with the help of "a" sparse coding method?
- "Specifically, the DA and HPO are regarded as dynamic schedulers, which adapt themselves to the update of network parameters and network architecture" is in fact a great summary to add to the abstract
- Organization of the paper (order of sections, figures etc needs to be verified again). For example, before reaching section 2, we have covered the motivation, some of the literature and results in Figure 1. We then start with another detailed literature review in section 2
- [Minor] - "the query based and the gradient based" - remove the
- [Minor] - "and then apply or search for" - rephrase sentence

(Ignoring grammatical errors from here on as there are many to list)

- For the experimental section, a discussion on choice of algorithms is required. IF AutoHAS is not similar to DHA in structure, joint optimization can be expected to get better results overall. Experimental results are also only marginally better, and the actual implementation of the competing algorithms is unclear apart from a mention of "All the methods are implemented by
ourselves. Dierent NAS algorithms for one dataset are performed under the similar parameter weights constrain to
ensure fair comparison." It is unclear how this fair comparison is actually made; a discussion on these details will ensure readers understand this claim

**Strengths And Weaknesses:**

## Strengths

+ Using the continuous parameters and a differential method to do joint DHA
+ Experimental results are promising when compared to other NAS methods.


## Weaknesses

- In general, grammar needs to be checked throughout
- Organization of the paper needs improvement
- Most choices that make continuous optimization possible are simplifying ones. For example, the data augmentation case, defined as choosing 2 consecutive augmentations, with parameters of each augmentation ranging from [0,10]. In practice, this is could be flexible. A discussion on this may help - "Can the formulation extend to multiple augmentations in any range?"
- Joint optimization is in fact $\it{not}$ joint optimization. It is a sequential update of different dimensions in order. The reasons for this as stated are two-fold - 1. Each parameter works differently, and $\theta$ or model parameters cannot be optimized directly, and 2. Intractable/ Impractical to optimize all parameters together. These are not strong reasons, and it is unclear if actual joint optimization may lead to even better results. Moreover, this seems like a justification to the final methodology chosen, and not like a motivation that lead to the method.

---

### Comment · Action_Editors · 2022-09-01
**To the Authors**

Dear Authors,

You currently have the opportunity to respond to the reviewers and address any issues that they raised in their reviews. I encourage you to take this opportunity to respond to their reviews.

Kind regards,

Action Editor

---

### Decision · Action_Editors · 2022-10-07

**Recommendation:** Accept with minor revision

**Comment:**

This paper proposes a method for jointly optimizing architecture, hyperparameters, and the data augmentation policy of a neural network in a joint fashion using a differentiable mapping for each component. This is an ambitious and difficult problem, and the method appears to work quite well, finding settings with high accuracy, low latency/FLOPS (even if this isn't explicitly accounted for, from what I can see), in a reasonably efficient manner. There are limitations, such as for discrete data, that should be discussed in the paper.

The discussion with the reviewers highlighted some points of clarification. Among them:

- How to generalize the augmentation optimizer to incorporate more augmentation strategies.
- The fairness of the experiments.
- A discussion on setting the hyperparameters of DHA.

Amongst others. There are also additional results on FLOPS/Latency of the discovered models.

Please incorporate these points from the discussions raised with the reviewers. Otherwise, the reviewers unanimously lean towards acceptance and feel that this is a solid step towards an end-to-end AutoML solution.

**Audience:**

Yes, this is relevant to the AutoML community.

**Claims And Evidence:**

Yes, the claims made in the submission are sufficiently supported by evidence. Namely, that DHA is competitive with contemporary approaches on several common benchmarks, and it does provide an end-to-end solution that optimizes architecture, hyperparameters, and data augmentation schemes in a joint fashion. There are limitations for discrete parameters, but this is not a claim of the paper.

---

> ### Author Response · Authors · 2022-10-10
> **Response to Action Editors**
>
> Thank you very much for your efforts! We will prepare the camera-ready version according to the suggestions. Code and video will also be provided.

---

> ### Author Response · Authors · 2022-10-26
> **Camera Ready Revision**
>
> Dear Action Editors and Reviewers,
>
> Thank you very much for your efforts! We have submitted the camera-ready revision of our paper. In the revision, we have
>
> 1. Discussed how to generalize the augmentation optimizer to incorporate more augmentation strategies in the experiments;
> 2. Provided the discussion about how to implement the baseline methods for a fair comparison;
> 3. Provided a discussion on setting the hyper-parameters in DHA in implementation details;
> 4. Added additional results on FLOPS/Latency of the discovered models in the supplementary materials.
>
> Please help to have a check. Thanks!
>
>
> Best Regards,
>
> Authors of DHA